ᵃ | **Open Peer Review** | Environmental Microbiology | Research Article
# Diversity and varying predation capacities of culturable Amoebozoae against opportunistic vibrios in contrasting Mediterranean coastal environments

Etienne Robino,[1] Angélique Perret,[1] Cyril Noel,[2] Philippe Haffner,[1] Laurent Intertaglia,[3,4] Marion Richard,[5] Noémie Descamps,[1] Axelle Sellier,[1] Laura Onillon,[1] Philippe Lebaron,[3,4] Delphine Destoumieux-Garzón,[1] Guillaume M. Charrière[1]

**ABSTRACT**  Free-living amoebae (FLA) are ubiquitous and can be found in many environments including soil, freshwater, and marine environments. They feed on various microorganisms and can play an important role in the food web and its dynamics. We previously described that FLA belonging to the *Vannella* genus isolated from oyster farms in the Thau lagoon in France are able to establish stable interactions with *Vibrionaceae,* suggesting they can play a role in pathogen dynamics. To further investigate the ecological interactions between FLA and Vibrionaceae in Mediterranean coastal waters, we conducted monthly sampling for 1 year at three contrasting sites. FLA populations were isolated from water and sediment samples on different bacterial lawns, including *Escherichia coli* or *Vibrio tasmaniensis*, *Vibrio crassostreae*, and *Vibrio harveyi*. Diversity analysis by v4-18S barcoding revealed distinct communities between fractions and sites. The Vannellidae were found significantly enriched in the water, whereas Paramoebidae were found enriched in the sediments. Additionally, uneven distribution of Vexilliferidae, Vermistellae, and, to a lesser extent, Acanthamoebidae and Subulatomonas contrasted between sampling sites. Selection of grazers on different bacterial lawns revealed that *V. tasmaniensis* inhibits the growth of most Vannellidae, whereas *V. crassostreae* inhibits the growth of Paramoebidae. These differences were further confirmed functionally using isolates belonging to each Amoebozoa taxonomic group. Overall, our results highlight the need for more comprehensive studies of the diversity and population dynamics of Amoebozoa in marine environments and indicate that the role of these diversified grazers in shaping *Vibrio* communities is complex and still poorly characterized.

**IMPORTANCE**  Although they can play an important role in shaping bacterial communities and as intracellular niches for potential pathogens, the diversity and ecology of free-living amoebae (FLA) in marine environments are still poorly characterized. Very few studies have used systematic approaches to unravel the population dynamics and ecological variations of FLA in different environments. Our study in marine coastal environments highlights that FLA taxonomic groups can harbor habitat preferences and have different predation capacities, suggesting that FLA-bacteria interactions need to be considered at different taxonomic levels to uncover generalist versus specialist adaptations.

**KEYWORDS**  free-living-amoeba, *Vibrio*, barcoding, ecology, predation, marine microbiology, bacteria-protozoa interactions

G razing protists, such as free-living amoebae (FLA), are found in many different environments. They are mainly found in freshwater and marine environments, but

**Peer Reviewers** Christopher Aaron Rice, Purdue University, West Lafayette, Indiana, USA; Marco Gabrielli, Eawag, Dübendorf, Switzerland

Address correspondence to Guillaume M. Charrière, guillaume.charriere@umontpellier.fr.

The authors declare no conflict of interest.

See the funding table on p. 15.

also in soil and air and can be associated with various hosts (1, 2). The diversity of FLA has been studied more extensively in freshwater environments than in marine environments, especially because human pathogenic amoebae are mostly found in freshwater (3). Amoebae are a polyphyletic group that branches along the eukaryotic tree and belongs to four supergroups, Amoebozoa, Excavata, Opisthokonta, and SAR (4). FLA feed on various microorganisms, including bacteria, yeasts, fungi, algae, or other protists, by phagocytosis and digest them in digestive vacuoles (5–8). As heterotrophic protists, they are central members of the food web in the environment, and their predation on bacteria is a major factor shaping microbial communities (9–13). Protist dynamics (e.g., abundance and diversity) are influenced by many environmental factors, such as prey type and abiotic factors including salinity, temperature, and oxygen availability (14–18). We observed previously that the diversity of FLA in the Mediterranean Thau lagoon near the oyster farming area seemed relatively low, with amoebae mainly belonging to the *Vannella* genus, and some of these *Vannella* could establish stable interactions with Vibrionaceae (19). Vibrios are γ-proteobacteria that live in aquatic environments, mostly in marine waters but can also be found in freshwater. They are ubiquitous and associated with many hosts from protozoa to metazoans, and these associations range from symbiosis to pathogenesis (20, 21). Vibriosis in marine animals can have devastating impacts in aquaculture, including oyster farming (22, 23). For example, they can be involved in mortality events of juvenile oysters *Crassostrea gigas* (24). Immunosuppressed oysters after viral infection are colonized by opportunistic *Vibrio* species that cause septicemia, leading to animal death. These opportunistic *Vibrio* species, which have been implicated in Pacific oyster mortality syndrome (POMS), belong to the species *Vibrio crassostreae*, *Vibrio tasmaniensis*, and *Vibrio harveyi* (25–27). Interestingly, cytotoxicity against hemocytes by *V. tasmaniensis* LGP32, *V. crassostreae* J2-9, and *V. harveyi* Th15_O_A01 has been shown to be a major driver of their virulence but relies on species-specific mechanisms (27, 28). The acquisition of such anti-eukaryotic activities could result from the long co-evolutionary history between bacteria and FLA. Some bacteria have evolved resistance to amoeba predation through extracellular and intracellular defense mechanisms that play a role in virulence against animal hosts (13, 29). Therefore, amoebae are considered evolutionary ancestors of interactions and act as a training ground for pathogenic bacteria by favoring the selection of virulence factors through coincidental evolution (30, 31). In Vibrionaceae, *Vibrio cholerae* has been shown to be able to survive in *Acanthamoeba castellanii* amoebae and to use some virulence genes that play a minor role in the interaction with the incidental human host (32). However, when *V. cholerae* leaves *Acanthamoeba castellanii* by exocytosis of the *Vibrio*-containing food vacuole, this can increase its virulence against mammalian hosts (33). In the case of *Vibrio vulnificus*, the virulence factor MARTX type III was shown to be involved in resistance to grazing by *Neoparamoeba permaquidensis* isolated from fish gills (34). By comparative cell biology, we have previously shown that the oyster opportunistic pathogen *V. tasmaniensis* LGP32 is able to resist phagocytosis by the environmental marine amoeba *Vannella* sp. AP1411 using some virulence factors also involved in oyster pathogenesis, in particular the secreted metalloprotease Vsm and the copper efflux pump CopA (19). Finally, lipopolysaccharide O-antigen variations in *Vibrio splendidus* strains were shown to determine resistance to grazing by *Vannella* sp. AP1411 (35). More recently, we described a natural association between *Vibrio bathopelagicus* of the Splendidus clade and *Paramoeba atlantica* (36). Taken together, these different reports suggest that amoeba-vibrio interactions are diverse and need to be further studied to better evaluate their role in *Vibrio* dynamics and pathogen emergence.

To study the ecology and interactions between FLA populations and *Vibrionaceae* in Mediterranean coastal waters, in the present report, we conducted monthly sampling for 1 year in three contrasting environments: first, in the Thau lagoon, which is used for oyster farming; second, near the port of Sète in the open-sea; and third, near the marine protected area of Banyuls-sur-Mer. FLA populations were isolated by culturing water and sediment samples on different bacterial lawns, including *Escherichia*

coli SBS363, *V. tasmaniensis* LGP32, *V. crassostreae* J2-9, and *V. harveyi* Th15_O_A01. Using 18S barcoding, we found that Amoebozoa dominated the FLA populations, with higher diversity in the sediment than in the water column, with distinct communities and site-specific diversity in the sediments. One of the most striking differences was that Vannellidae are abundant and specifically enriched in the water column, whereas Paramoebidae are abundant and specifically enriched in the sediments. Furthermore, we show that these two families of amoebozoans have contrasting predation capacities against two different species of *Vibrio* of the Splendidus clade, namely, *V. tasmaniensis* LGP32 and *V. crassostreae* J2-9. This was further confirmed functionally using amoeba isolates from each family. Overall, our results show that the nature of amoeba-*Vibrio* interactions may occur at very different taxonomic levels and can play a role in the dynamics of opportunistic pathogens in coastal marine waters. They highlight that the diversity of Amoebozoa is still understudied and needs to be further characterized in different marine environments.

## MATERIALS AND METHODS

### Bacterial strains, amoeba strain, and growth conditions

*E. coli* strain SBS363 was grown in Luria-Bertani (LB) or LB agar (LBA) at 37°C for 24 hours prior to experiments. *V. tasmaniensis* LGP32, *V. crassostreae* J2-9, and *V. harveyi* Th15_O_A01 were grown in LB + NaCl 0.5 M or LBA + NaCl 0.5 M at 20°C for 24 hours prior to experiments. *Vibrio* strains carrying the pMRB-GFP plasmid were grown in LB + NaCl 0.5 M supplemented with chloramphenicol (10 $\mu g.mL^{-1}$) at 20°C for 24 hours prior to experiments. *Vannella* sp. AP1411 (isolated in a previous study [19]) and *Paramoeba atlantica* strain CCAP1560/9 (purchased from the CCAP collection, Scotland, UK) were cultured at 18°C–20°C in 3 mL of 70% sterile seawater (SSW) supplemented with 200 $\mu$L of an *Escherichia coli* SBS363 suspension ($OD_{600}$ = 20).

### Sampling

Sampling was performed monthly for 1 year between 2017 and 2018 in three contrasting environments in southern France. Water and sediment were collected in the Thau lagoon adjacent to oyster beds at the Bouzigues station Ifremer-REPHY (GPS: N 43°26'.058" – E 03°39'.878"), in the open sea near the port of Sète (GPS: N 43°23'.539"– E 03°41'.933"), and near the Banyuls-sur-Mer marine reserve at the SOLA station (GPS: N 42°29'.300" – E 03°08'.700") (Fig. S1). The surface water (1 m depth) was sampled with a Hydrobios bottle at the three sites, and the first centimeters of sediments were sampled with cores taken by divers at 9, 10, and 30 m depths, respectively, at Thau lagoon, Sète, and Banyuls-sur-Mer. The sediments from Thau and Sète were muddy and sandy, respectively, while the sediments from Banyuls-sur-Mer had a mixed composition. Water was filtered on the boat using a 63 μm pore-size nylon filter and then refiltered in the laboratory using a 5 μm pore size MF-Millipore membrane. The 5 μm pore size membrane was then cut into four pieces, and three quarters were placed upside down on a lawn of *E. coli* SBS363 seeded on 70% SSW agar. For the sediment samples, one gram was placed in the center of a lawn of *E. coli* SBS363 seeded on 70% SSW agar in triplicate and incubated at 20°C. After 2 weeks, the FLA that grew and colonized the area outward of the initial sample (filter piece or sediment) were gently recovered over the entire surface of the agar plate (avoiding the initial sample) by using 2 mL of 70% SSW and gently scraping the top of the agar with a microbiology spreader, 1 mL of the FLA suspension was cryopreserved at −80°C for each condition. The same steps were performed with *V. tasmaniensis* LGP32, *V. crassostreae* J2-9, and *V. harveyi* Th15_O_A01 in May and October 2017 during oyster mortality events and in January and February without mortality events. A total of 76 samples from *E. coli* plates and 84 samples from vibrio plates were collected for diversity analysis using 18S barcoding.

## Amoebozoa diversity analysis on different nutritive sources

Total DNA from the frozen FLA suspensions was extracted using a Blood and Tissue kit from Marcherey Nagel with an adapted initial lysis phase using 0.1 mm zirconium beads under string agitation in T1 buffer. Eukaryote diversity was analyzed by barcoding based on the polymorphism of the v4 loop of the 18S rRNA coding gene. PCR was performed using universal primers at an annealing temperature of 53°C (TAReuk454FWD-illumina: 5′-TCG TCG GCA GCG TCA GAT GTG TAT AAG AGA CAG YRC CAG CAS CYG CGG TAA TTCC-3′ and TAReukRev3 Illumina: 5′-GTC TCG TGG GCT CGG AGA TGT GTA TAA GAG ACA GYR ACT TTC GTT CTT GAT YRA-3′) (37). Paired-end sequencing (300 bp read length) was performed using the GenSeq platform (Labex CEMEB, Montpellier, France) on the MiSeq system (Illumina). Raw data were then processed using the SAMBA v.3.0.1 pipeline (https://gitlab.ifremer.fr/bioinfo/workflows/samba), a standardized and automated workflow for meta-barcoding analyses. This workflow, developed by SeBiMER (Ifremer's Bioinformatics Core Facility), is an open-source modular workflow for processing eDNA metabarcoding data. SAMBA was developed using the Nextflow workflow manager (38) and allows the execution of three main components: data integrity verification, bioinformatics processes, and statistical analyses.

Seventy-two samples were obtained from *E. coli* agar plates; six samples that yielded less than 1,000 sequences were removed (10-17-Ba-S-E-coli, 11-17-Th-S-E-coli, 06-17-Th-S-E-coli, 09-17-Th-S-E-coli, 12-17-Ba-S-E-coli, and 11-17-Ba-S-E-coli), and two samples were considered outliers due to a species composition that was completely different from the rest of the samples (08-17-Th-W-E-coli and 09-17-Ba-W-E-coli). Analysis of the remaining 64 samples resulted in a total of 2,780,405 sequences after data integrity checks. QIIME 2 was used for primer removal using the Cutadapt plugin, sequence quality check was performed using the DADA2 R package (removal of low-quality sequences, assembly of forward and reverse sequences, and removal of chimeras), and amplicon sequence variant (ASV) clustering was done using the dbOTU3 algorithm to obtain a total of 2,556 ASVs (39–42). PR2 database was used for taxonomic assignment (version 4.13.0) with taxonomy filtering to select only ASVs belonging to the phylum of Amoebozoa, which consists only of FLA (43). A total of 465 ASVs were regrouped into a table containing the names of the ASVs with their taxonomic affiliations, as well as the names of the samples containing the ASVs and number of reads identified for each sample (Table S1).

For the analysis of the samples acquired on the four different nutrient sources (*E. coli* SBS363, *V. harveyi* Th15_O_A01, *V. tasmaniensis* LGP32, and *V. crassostreae* J2-9), 112 samples were obtained (i.e., 84 samples from *Vibrio* lawns and 28 samples from *E. coli* lawns); 6 samples with less than 900 sequences were removed (02-18-Ba-S-LGP32, 01-18-Th-S-A01, 10-17-Ba-S-E-coli, 05-17-Th-S-LGP32, 05-17-Ba-S-J2-9, and 05-17-Se-S-J2-9). Analysis of the 106 remaining samples resulted in a total of 5,298,844 sequences after data integrity checks. A total of 3,084 ASVs were obtained, and a total of 474 ASVs were affiliated to the phylum of Amoebozoa (see Table S5). SAMBA then performed extensive analyses of alpha and beta diversity using custom R scripts (44). Alpha diversity was examined using the Chao1, Shannon, and InvSimpson indices. Beta diversity was investigated using principal coordinate analysis (PCoA) with DESeq2 normalization method and UniFrac distance matrix. Rarefaction curves revealed a high variability in sequencing depth among samples (Fig. S5), leading to a substantial loss of information when subsampling to a uniform depth, hence DESeq2 normalization was preferred to preserve more of the underlying community structure and to provide more stable diversity estimates across replicates. When needed, a permutation test for multivariate dispersion was performed to evaluate dispersion of the samples. Additionally, ANCOM analysis on abundance data and indicspecies (from Vegan package) on relative abundance data were used to identify taxa that are specific to sampling sites or fractions. Repartition of the specific and shared ASVs across the different conditions highlighted the taxonomy for each variable or combined variables and was plotted using UpsetR v.1.4.0. The phylogenetic tree was performed using MAFFT and FastTree

(maximum likelihood tree) and annotated using iTOL software, highlighting fraction, sites, and season variables (45).

## Grazing assay

To prepare the co-culture of vibrios and amoebae, 1 mL of *Vibrio* overnight culture ($3.10^9$ bacteria/mL) was mixed with 100 µL of 3-day-old *Vannella* sp. AP1411 or *Paramoeba atlantica* culture ($5.10^5$ cells/mL) or with 100 µL of 70% SSW for the control condition. A volume of 50 µL per well of the mixed culture was seeded on 500 µL of 1% SSW agar in a 24-well plate with a transparent flat bottom. Amoebae and bacteria were carefully homogenized in the wells, allowed to dry by incubating for 4 hours at room temperature under a sterile hood, and then incubated at 18°C in a humidified atmosphere. GFP fluorescence intensity was measured daily for 7 days using a TECAN plate reader (λex 480 nm/λem 520 nm). To estimate the effect of amoeba grazing activity on the abundance of live GFP-expressing vibrios, the fluorescence intensity of wells containing amoebae was compared to the fluorescence of vibrios in the absence of amoebae and expressed as a ratio. Each condition was performed in triplicate, and the results shown are representative of three independent experiments. Error bars represent the standard error of the mean (± SEM). Statistical analysis was performed using RM-ANOVA on the independent experiments.

## RESULTS

### Amoebozoa diversity varies greatly between the water column and sediments

To describe the diversity and distribution of cultivated grazers in Mediterranean coastal environments, we performed a monthly sampling survey of three contrasting environments. The Thau lagoon is used for oyster farming, where oyster mortalities due to POMS occur annually. It represents a semi-enclosed coastal system with restricted seawater renewal and substantial freshwater inputs from watersheds, resulting in salinity variations (46). The Mediterranean Sea, near the port of Sète, is an open-sea area without aquaculture activities. The marine protected area of Banyuls-sur-Mer, which is more distant from the two other sites, is characterized by restricted human activities to protect the marine biodiversity. Monthly sampling was carried out over an entire year in the three chosen sites (Fig. S1). Sediments and water were sampled every month for a whole year, from March 2017 to March 2018. To isolate cultivable amoebae, samples were placed on non-nutritive plates with a layer of *E. coli* SBS363 as a non-pathogenic and permissive nutritive food source to isolate the highest diversity possible under these conditions. After 15 days of incubation at 20°C, total cultivable diversity was estimated by sequencing the v4 hypervariable region of 18S rRNA using universal primers (37). We then used the SAMBA pipeline to process the sequencing data (see Materials and Methods). Due to the lack of taxonomic references in 18S SSU databases for some protist groups, we focused on ASVs belonging to Amoebozoa, as they represent the majority of ASVs with a taxonomic affiliation, and this taxonomic group contains only FLA. Initial analyses showed that the alpha-diversity and beta-diversity of Amoebozoa differed significantly between sediment and water column samples, independent of sampling location (Fig. 1A and B). The Chao1 index (estimator of species richness) was significantly different, with a median Chao1 in the water column of 6.5 ASVs compared to a median of 18 ASVs for the sediments, indicating that alpha diversity tends to be lower in the water column than in the sediments (ANOVA test; $P = 0.0138$; Table S2). The other two alpha diversity indices, Shannon and InvSimpson, were not significantly different, indicating that the differences between the fractions were mostly due to low abundance species. For beta-diversity, PCoA revealed different Amoebozoa communities between water and sediment samples (Adonis test; Adonis $R^2 = 0.19$, $P = 0.0001$; Table S3; Fig. 1A). In addition, sediment samples were significantly more grouped than water samples (permutation test for dispersion, $P$-value = 0.037), suggesting a more stable

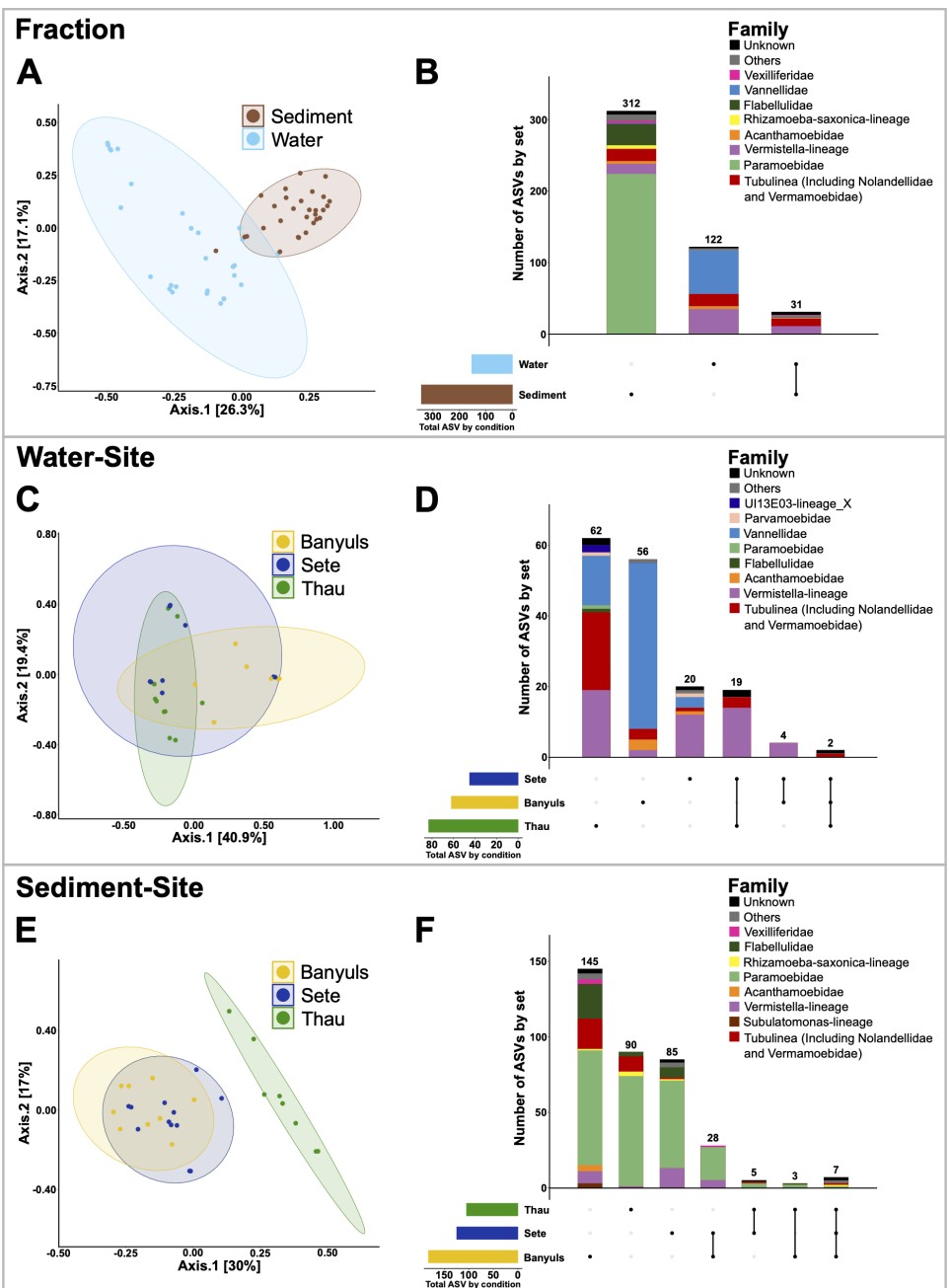

**FIG 1** Amoebozoa diversity is structured according to sampling fraction and sampling site. PCoA of the unweighted UniFrac matrix distance revealed that the beta diversity of Amoebozoa in the sediment is different from the beta diversity in the water samples (A). The distribution of ASVs showed that ASVs belonging to Paramoebidae are particularly enriched in the sediment, whereas ASVs belonging to Vannellidae are particularly enriched in the water samples (B). PCoA of the unweighted UniFrac matrix distance between water samples from the different sampling sites showed that the beta-diversity of Amoebozoa in water is highly variable and does not differ significantly between sampling sites (C). ASV distribution shows that most ASVs found in water samples are specific to each site and the community structure is variable, but ASVs of each family are found in all sites (D). The PCoA of the unweighted UniFrac matrix distance between the sediment samples from different sampling sites showed that the beta diversity of Amoebozoa in the sediments is more homogeneous, and the beta diversity found in the Thau sediments is different from the beta diversity in the sediments of Sète and Banyuls, which are more similar (E). The distribution of ASVs shows that most ASVs found in sediments are specific to each site, and the community structure is variable, but ASVs of each family are found in all sites (F).

community over the sampling months (Fig. 1A). The repartition of ASVs grouped by fractions showed that of the 465 ASVs, 312 ASVs (67.1%) were found in sediments only, while 122 ASVs (26.2%) were found in water only (Fig. 1B). In addition, very few ASVs were shared by both fractions (31 ASVs, representing 6.7% of the total ASVs). This distribution of ASVs illustrates the statistical differences found in the beta-diversity analyses, highlighting a higher richness in the sediments than in the water samples. Consistent with the differences in beta diversity observed between the two different habitats (water column versus sediments), the distribution of two abundant Amoebozoa families appeared particularly contrasted between water samples and sediments. Paramoebidae were found specifically enriched in the sediments, whereas Vannellidae were found specifically enriched in the water column (ANCOM analyses; Table S4).

## The diversity of cultivable marine Amoebozoa depends on the geographical location

Secondly, we investigated the differences in Amoebozoa diversity between the three contrasting sampling sites, Banyuls-sur-Mer, Sète, and Thau. No significant differences in alpha diversity were observed between the sites, but beta diversity using PCoA with the DESeq2 normalization method and UniFrac distance matrix showed that the beta diversity of each site was specific and significantly different from the others (Table S3). Due to the strong fraction effect, the taxonomic level of beta-diversity differences between sites was difficult to evaluate independently of fraction. Therefore, we compared the beta-diversity of Amoebozoa between sites in the water samples on the one hand and in the sediments on the other hand. Although the communities were all significantly different (Table S3), the three groups of samples overlapped (Fig. 1C and D). These results suggest that the variance between samples belonging to the same site was large and in a similar range for each site. To identify taxa with uneven distribution between sampling sites, ANCOM analyses as well as indicspecies (Vegan package) on relative abundances were performed. ANCOM analysis suggested that 12 taxa are unevenly distributed between water samples from the different sites (Table S4), however with a low statistical power due to low abundances for most of them. Accordingly, indicspecies analysis confirmed that only Vannellidae were significantly enriched in water samples from Banyuls when considering relative abundances (Fig. S2). For the sediment samples, PCoA and UniFrac distance matrix showed that the beta diversity was significantly different between each site, with slightly more homogeneity between samples from the same site than what was observed for the water samples (Fig. 1E and F). The group of samples from Banyuls-sur-Mer and Sète overlapped, while the group of samples from Thau was very distinct, suggesting that the Amoebozoa communities in the sediments from Banyuls-sur-Mer and Sète are more similar to each other than to the community from Thau. ANCOM analyses suggested that Vexilliferidae and Vermistella lineage harbor an uneven distribution between sediment samples from the different sites. Relative abundance analyses with indicspecies confirmed that Vexilliferidae were more abundant in samples from Banyuls, whereas Vermistellae were more abundant in samples from Sète. Moreover, this analysis revealed that Acanthamoebidae and Subulatomonas lineage were significantly enriched in samples from Banyuls, although the absolute abundances were very low (Table S4; Fig. S2). Overall, the ASV distribution suggested additional intra-family variations. The number of ASVs shared between sites in the water samples or sediment samples was low (Fig. 1D and F), but these differences were not found statistically significant due to the high heterogeneity between samples from the same site. Taken together, these results indicate that Amoebozoa diversity is very heterogeneous and variable. Site-specific assemblages of the Amoebozoa community was more clearly evidenced in the sediment than in the water fraction.

## Seasonal variations in Amoebozoa diversity occur at the subgenus level

We next attempted to evaluate seasonal variations in Amoebozoa diversity with respect to both fractions and the three sites. No significant differences were found for either

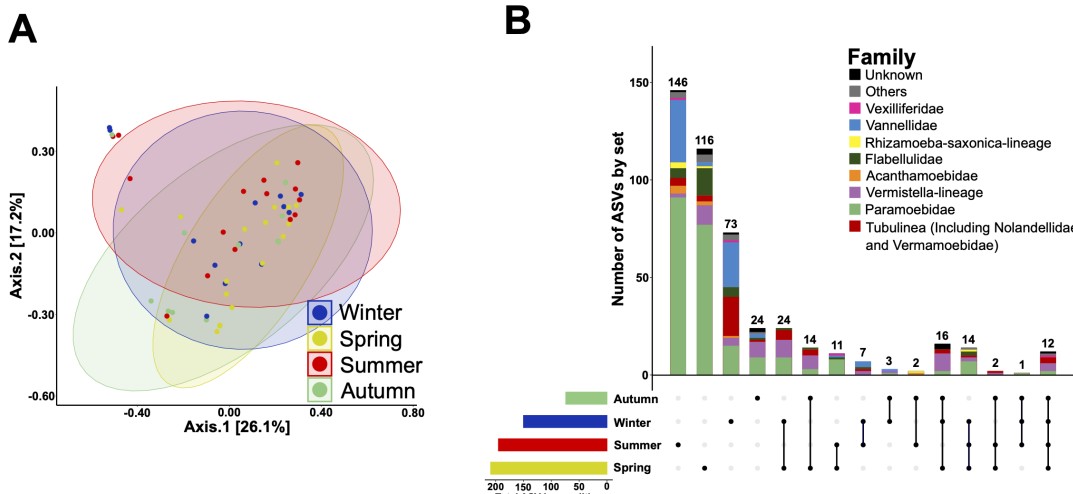

**FIG 2** Seasonal variation of Amoebozoa diversity is highly heterogeneous, but more ASVs are found in spring and summer than in winter and autumn. PCoA of unweighted UniFrac matrix distance revealed that the beta diversity of Amoebozoa at different seasons is highly variable and overlaps between seasons (A). The distribution of ASVs showed that the total number of different ASVs is higher in spring and summer than in winter and autumn, and ASVs belonging to each family are found in all seasons in varying numbers (B).

alpha or beta diversity (Fig. 2A; Tables S2 and S3). ASVs belonging to the majority of the FLA taxa were found at all seasons, and statistical analyses did not allow highlighting specific uneven distribution between seasons (Fig. 2B). However, the number of different ASVs belonging to the same family between seasons varied noticeably. A higher number of ASVs differed in summer and spring than in winter and fall. Among the 465 identified ASVs, 146 (31.4%) and 116 (24.9%) were specific to summer and spring, respectively, while only 73 (15.7%) and 24 (5.2%) were specific to winter and fall, respectively (Fig. 2B). Such variations suggested subgenus-level diversity variations associated with seasonality, and these taxonomic variations were apparent in the phylogenetic classification of ASVs, with some intra-genus clades being particularly present in a particular season (Fig. 3). For example, among Paramoebidae, ASVs belonging to clade 11 were found only in summer in Thau lagoon, whereas among Vannellidae, ASVs belonging to clade 2 were found in winter in Thau lagoon, and ASVs belonging to clades 1 and 3 were found in summer in Banyuls (Fig. 3). All these results suggest the presence of contrasting seasonal variations at the subgenus level, but our data set did not allow us to highlight statistically significant seasonal variations of Amoebozoa diversity and abundance due to the high heterogeneity between samples.

## Different Amoebozoa taxonomic groups have varying predation capacities against different pathogenic *Vibrio* strains

The diversity of amoeba-*Vibrio* interactions and their specificity remains poorly studied. We wondered here whether different pathogenic *Vibrio* strains with different anti-eukaryotic virulence mechanisms would select for different predators. Therefore, during four different months of the sampling campaign, *V. tasmaniensis* LGP32, *V. crassostreae* J2-9, and *V. harveyi* Th15_O_A01 were used as food sources in addition to the regular *E. coli* SBS363 (Fig. S1). Since the abundance of pathogenic *Vibrio* in the two different fractions, water column and sediments, differs during oyster mortality (47), we chose 2 months during oyster mortality events (May 2017 and October 2017) and 2 months without mortality events (January 2018 and February 2018). The highest contrast in alpha diversity of Amoebozoa was observed between *V. harveyi* Th15_O_A01 plates and *V. tasmaniensis* LGP32 plates, as shown by all alpha diversity indices (Table S6), while *E. coli* SBS363 and *V. crassostrea*e J2-9 plates showed intermediate alpha diversity. Beta diversity did not show significant differences between the four prey species (Fig. 4A

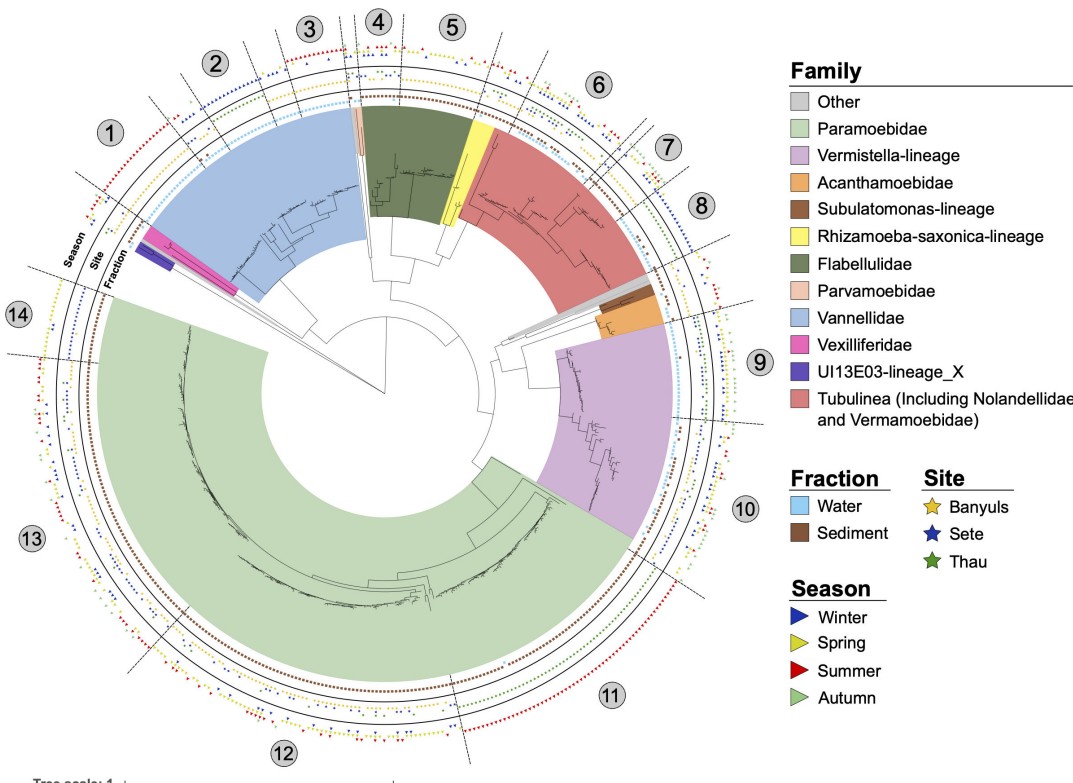

**FIG 3** Phylogeny of ASVs highlighting intrafamilial clades of Amoebozoa present in different habitats. The phylogenetic classification of ASVs was performed using MAFFT and FastTree (maximum likelihood tree) and annotated using iTOL software, highlighting fraction, location, and season variables. Numbers highlight different phylogenetic clades that can be delineated based on phylogenetic distance and environmental variables.

and Table S6). The repartition of ASVs confirmed the results of alpha and beta diversity. Indeed, we identified the maximum number of specific ASVs and the maximum diversity with *V. harveyi* Th15_O_A01 plates (140 ASVs representing 29.5% of the 474 total ASVs), with specific ASVs belonging to the Paramoebidae family dominating the observed diversity and contrasting between the four different prey species (Fig. 4B). Of all detected ASVs, 58.7% were found on only one type of bacterial lawn, while 41.3% were found on at least two different types of bacterial lawn. These 58.7% of ASVs found only on one bacterial lawn had very low abundance, which explains why there was no significant difference observed in beta-diversity between bacterial lawns overall. Interestingly, 39 ASVs, representing 8.2% of the total ASVs and belonging to different Amoebozoa families, had the ability to feed and grow on all four different types of bacterial lawns, suggesting a less specific predatory activity (Fig. 4B).

## *V. tasmaniensis* LGP32 strongly inhibits the growth of Vannellidae

Since the two most contrasting families appeared to be the Paramoebidae and Vannellidae in terms of sampling fraction, sampling sites, and predatory capacity, we analyzed these two families in more detail. Phylogenetic classification of Vannellidae ASVs revealed three distinct clades with different characteristics. Clade 1 was composed of relatively distant ASVs identified on the four bacterial lawns, mainly from the water samples, with three ASVs from sediments at the three sites during the four seasons (Fig. 5). Clade 2 gathered more closely related ASVs found mainly on the *E. coli* SBS363 lawn, with two ASVs and four ASVs from the *V. tasmaniensis* LGP32 and *V. harveyi* Th15_O_A01 lawns, only in the water column at Thau and during February (Fig. 5). Clade 3 was composed of ASVs found mainly on *V. harveyi* Th15_O_A01 and *V. crassostreae* J2-9 lawns, with some ASVs found on *E. coli* SBS363 lawn, only in the water column and at

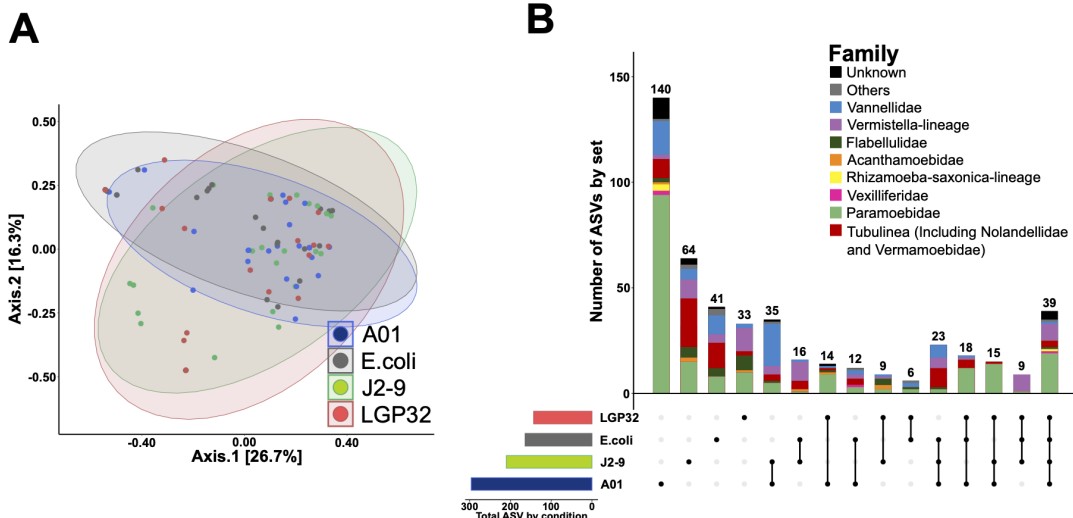

**FIG 4** Selection on different bacterial prey reveals intra-family variation in the predatory capacity of Amoebozoa against different opportunistic vibrios. PCoA of the unweighted UniFrac matrix distance reveal that the beta diversity of Amoebozoa growing on the different bacterial prey is highly variable and mostly overlaps between conditions (A). The distribution of ASVs reveals that the total number of different ASVs is higher on *V. harveyi* Th15_O_A01 compared to the three other bacterial lawns, but 39 ASVs belonging to the different Amoebozoa families were able to grow on all four bacterial lawns (B).

Banyuls-sur-Mer, and mainly during January and February, with two ASVs identified during May (Fig. 5). These observations illustrate well the existence of geographical and seasonal specificities in the phylogenetic diversity of Vannellidae. In addition, amoebae belonging to the Vannellidae appeared to be much less prone to grow on the *V. tasmaniensis* LGP32 lawn than on the other bacterial lawns; only a few ASVs were identified from samples grown on the *V. tasmaniensis* LGP32 lawn. These results are reminiscent of our previous study showing that *V. tasmaniensis* LGP32 is resistant to predation by the amoeba *Vannella* sp. AP1411 (19).

## The Paramoebidae represent a highly diversified taxonomic group with diverse predatory capabilities

The Paramoebidae represented the amoeboid family with the highest number of identified ASVs (Fig. 6). Furthermore, this family appeared to contain ASVs with some having a tendency toward generalist predatory capacity and some others appearing to be more specialized, but most of them showed a high capacity to feed on *V. harveyi* Th15_O_A01. There were two clades (clades 3 and 5) able to feed on the four food sources, mainly found in the sediments, at the three sites for clade 3, with a majority at Banyuls-sur-Mer and Sète for clade 5, during the 4 months sampled with fewer ASVs in October. In addition, four more specialized clades (clades 1, 2, 4, and 6) were identified, showing some differences. Clades 1 and 4 were found similarly in the water column, at Banyuls-sur-Mer and in October. These two clades are interesting because very few of the total ASVs were identified during October. Clades 2 and 6 were mainly identified at Sète and during May. However, ASVs of clade 2 were mainly found in the water column and clade 6 in the sediments. These results highlight that Paramoebidae - the largest family of Amoebozoa sampled here - is highly diverse in terms of ecological dynamics and predatory capacity.

## Contrasting predation capacities against phylogenetically related *Vibrio* strains of the Splendidus clade are functionally validated using Vannellidae and Paramoebidae isolates

To further confirm that Vannellidae growth was inhibited by *V. tasmaniensis* LGP32 and Paramoebidae growth was inhibited by *V. crassostreae* J2-9, we first compared the

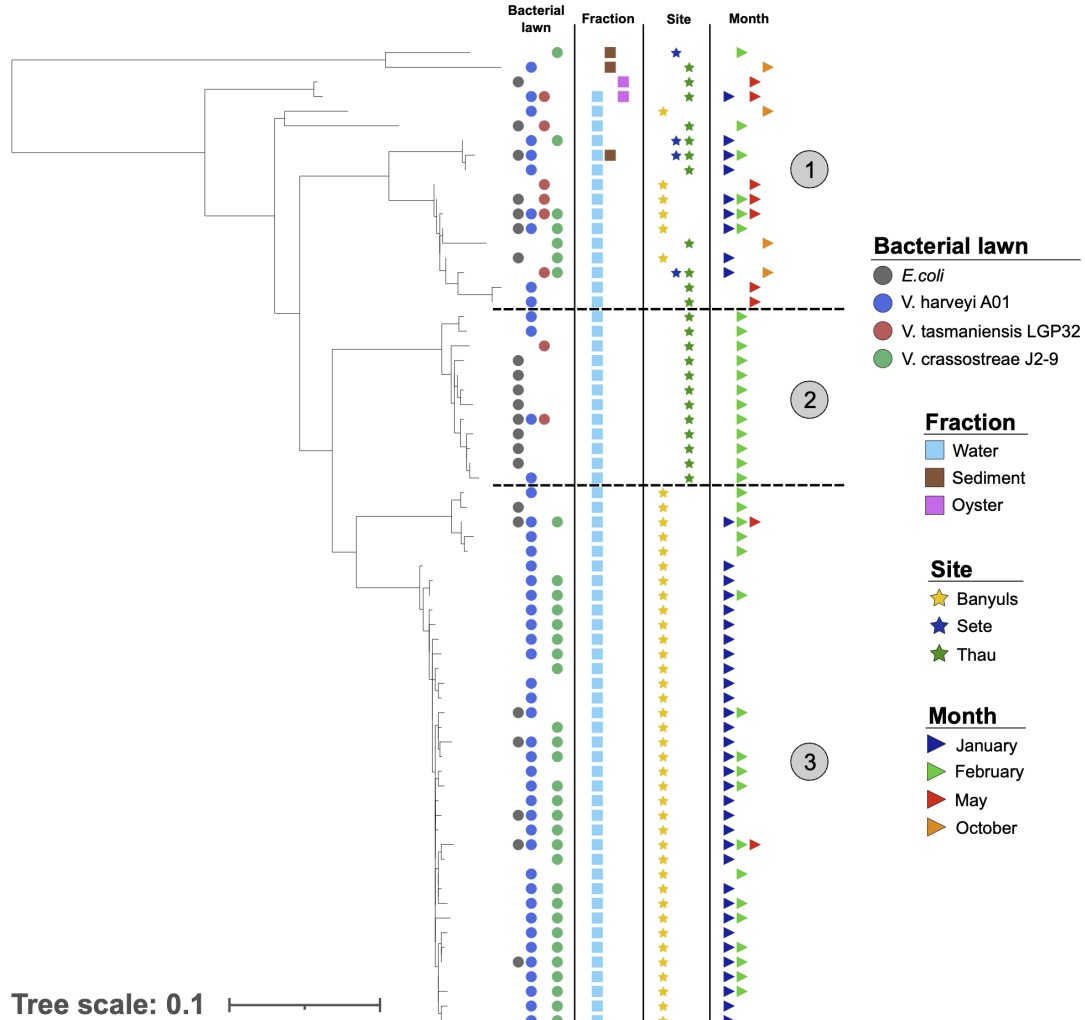

**FIG 5** Phylogeny of ASVs highlighting clades of Vannellidae found in different environments and with different predatory capacities. The phylogenetic classification of ASVs was performed using FastTree (maximum likelihood tree) and annotated using iTOL software, highlighting fraction, location, and season variables. Clade 1 was heterogeneous and found in different conditions but was able to grow on any of the bacterial prey. Clade 2 was found in Thau in February and grew mainly on *E. coli*, and clade 3 was found mainly in Banyuls in January and February and was unable to grow on LGP32.

relative abundance of these two families of Amoebozoa across all samples cultured on *V. tasmaniensis* LGP32 and *V. crassostreae* J2-9 lawns. The relative abundance of Paramoebidae was found to be significantly higher than that of Vannellidae on *V. tasmaniensis* LGP32 lawn (Fisher's test, *P* < 0.05; Fig. 7A). On the contrary, the relative abundance of ASVs belonging to Paramoebidae on *V. crassostreae* J2-9 lawn was found to be significantly lower than the abundance of ASVs belonging to Vannellidae (Fisher's test, *P* < 0.05). These results suggest that *V. tasmaniensis* LGP32 inhibits the growth of Vannellidae, whereas *V. crassostreae* J2-9 tends to inhibit the growth of Paramoebidae. To functionally validate these results, we performed grazing experiments with *V. tasmaniensis* LGP32-GFP and *V. crassostreae* J2-9-GFP strains to quantify the predation activity of *Vannella* sp. AP1411 and *Paramoeba atlantica* (strain CCAP1560/9) over time. *Vannella* sp. AP1411 was unable to graze on *V. tasmaniensis* LGP32 (as previously published by Robino et al. [19]), whereas *V. crassostreae* J2-9 was rapidly grazed, as shown by the rapid decrease in relative fluorescence of GFP, which was reduced to a ratio of 0.4 after 5 days (ANOVA, *P* < 0.001; Fig. 7B). In contrast, grazing experiments with *Paramoeba atlantica* showed that *V. tasmaniensis* LGP32 was grazed faster than *V. crassostreae* J2-9, although the differences between the two *Vibrio* strains were less pronounced than with *Vannella* sp. AP1411

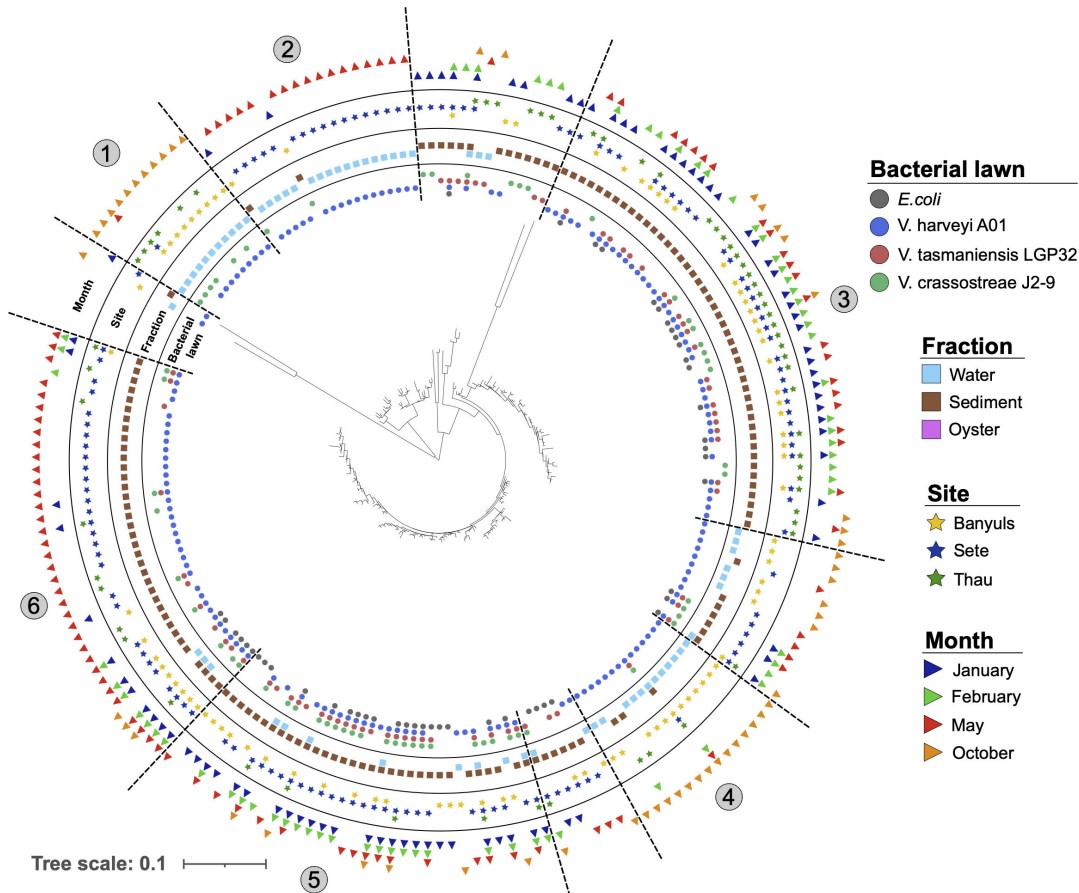

**Bacterial lawn**
- ● *E.coli*
- ● V. harveyi A01
- ● V. tasmaniensis LGP32
- ● V. crassostreae J2-9

**Fraction**
- ■ Water
- ■ Sediment
- ■ Oyster

**Site**
- ★ Banyuls
- ★ Sete
- ★ Thau

**Month**
- ▶ January
- ▶ February
- ▶ May
- ▶ October

Tree scale: 0.1

**FIG 6** Phylogeny of ASVs highlighting different clades of Paramoebidae found in different environments with different predation capacities. The phylogenetic classification of ASVs was performed using MAFFT and FastTree (maximum-likelihood tree) and annotated using iTOL software, highlighting fraction, site, and season variables. Some clades were ubiquitous and mostly generalist, as they could grow on the four bacterial lawns, such as clades 3 and 5, whereas other clades appeared to have more restricted habitats with a more limited predation capacity.

(ANOVA, $P < 0.05$; Fig. 7C). Taken together, these functional data confirm that the two genera, Vannellidae and Paramoebidae, have different predatory capacities against different pathogenic *Vibrio* strains belonging to the Splendidus clade.

## DISCUSSION

To investigate the ecology of FLA populations and their interactions with *Vibrionaceae* in the Mediterranean coastal environment, we conducted monthly sampling for 1 year in three contrasting habitats. FLA populations were isolated by culturing water and sediment samples on different bacterial lawns, including *E. coli* SBS363, *V. tasmaniensis* LGP32, *V. crassostreae* J2-9, and *V. harveyi* Th15_O_A01. Analysis of protist diversity in the different samples by v4-18S rRNA barcoding revealed distinct communities of Amoebozoa between the sediments and the water column, with Vannellidae significantly enriched in the water column and Paramoebidae significantly enriched in the sediments. Moreover, the diversity of Amoebozoa in the sediments was more specific to the sampling sites than in the water column. Selection of grazers on different bacterial lawns revealed that *V. tasmaniensis* LGP32 inhibited the growth of most Vannellidae, whereas *V. crassostreae* J2-9 tended to inhibit the growth of Paramoebidae. These differences were further confirmed in functional grazing assays using isolates belonging to each Amoebozoa taxonomic group. Overall, our results highlight the need for more comprehensive studies of amoebae diversity and population dynamics in marine waters, and the role of

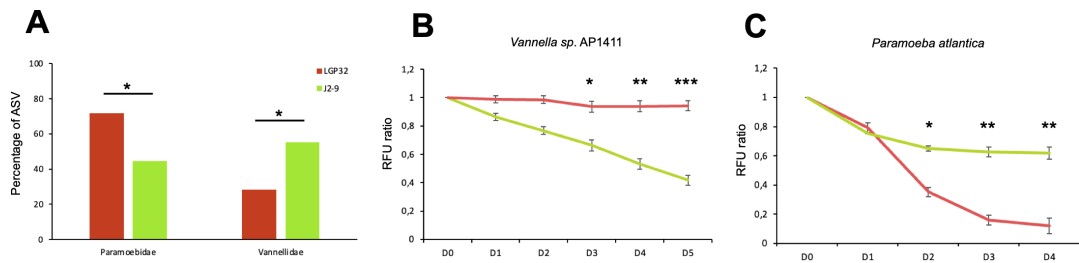

**FIG 7** *V. tasmaniensis* LGP32 and *V. crassostreae* J2-9 show contrasting resistance to predation by Vannellidae and Paramoebidae. The abundance of ASVs belonging to Paramoebidae or Vannellidae growing on LGP32 or J2-9 turf was significantly different (Fisher's test; $P < 0.05$) (A). Grazing experiments with *Vannella* sp. AP1411 showed that J2-9 was more sensitive than LGP32 (one experiment representative of three independent experiments, RM-ANOVA test; $P < 0.001$). (B) Grazing experiments with Paramoeba atlantica showed that J2-9 was more resistant than LGP32 (one experiment representative of three independent experiments, RM-ANOVA test with post-hoc test for pairwise comparison; $*P < 0.05$, $**P < 0.01$, $***P < 0.001$) (C).

these diversified grazers in shaping *Vibrio* communities and pathogen dynamics remains poorly characterized.

In contrast to our previous observations, by conducting a more extensive field survey, we found, in addition to Vannellidae, many other Amoebozoa belonging to Paramoebidae, Tubulinea (regrouping Vermamoebidae and Nollandelidae), Rhizamoeba-saxonica, Vermistellae, Flabellulidae, Vexilliferidae, and Parvamoebidae families. Although some were more abundant than others, including Paramoebidae, Vannellidae, Vermistellae, Tubulineae, and Flabellulidae, most had been previously reported from various marine environments (2, 48–53). A limitation of our study was the use of universal v4-18S primers to analyze FLA communities. FLA are very diverse and paraphyletic, and the use of universal v4-18S primers may have precluded identifying the full diversity, especially in taxonomic phylum other than Amoebozoa, such as Heterolobosea, for example (54). Besides Amoebozoa, the other abundant phylum of grazers detected here was Cercozoa, but the taxonomic assignment was limited due to the lack of reference sequences in 18S databases. Even for Amoebozoa, taxonomic assignment based on 18S rRNA partial sequences can be inconsistent at the species level and in some cases even at the genus level, which is the reason why we analyzed the diversity of Amoebozoa mostly at the family level.

A limited number of studies have performed a systematic sampling survey over time to compare the diversity of FLA between sites and between different fractions in marine environments as reported here. Overall, the higher diversity of Amoebozoa observed in sediments compared to water samples is consistent with their benthic grazer lifestyle. Distinct grazer communities between sediments and water columns have also been reported by other studies that found different protist communities between surface and deep waters of the same site as well as between different sediment depths (17, 55, 56). Here, several ASVs belonging to the different families were found in the different environments, suggesting that most of the diversity differences appear to be at the intrafamily and probably intrageneric level. Nevertheless, a limited number of ASVs were found in all sampling sites, suggesting that cosmopolitan strains of Amoebozoa are rare. However, the distribution of ASVs between the two different habitats was particularly contrasting for the Vannelliadae, which were specifically enriched in the water column, and the Paramoebidae, which were specifically enriched in the sediments. This may be related to a specific lifestyle and adaptations, as Vannellidae are known to form characteristic elongated filopodia in their planktonic form, which may provide an advantage for swimming and movement using water currents (57). In contrast, Paramoebidae are frequently reported in marine sediments and are rare in freshwaters, suggesting a specialized marine benthic lifestyle (52, 58–60). Amoebozoa communities in the sediments were found here to be more stable over the sampling months but contrasting between sites, whereas communities from the water samples were found to be more variable and less contrasted between sampling sites. This may be explained by differences in the physical characteristics and variability of the two fractions. The

water column is subject to significant variability and rapid mixing by water movement and currents, increasing the connectivity between environments, in contrast to the sediments, which are more preserved from mixing events. In addition, the physicochemical composition of sediments between sites (e.g., depth, sand/mud composition, and oxygen concentrations) is likely to be different, which could have a strong influence on the niche characteristics and composition of microorganism communities (16, 56). Further efforts in exhaustive comparative analyses between contrasting environments are still needed to unravel the most important environmental factors shaping FLA communities.

Unfortunately, the analysis of seasonal variation was hampered by the heterogeneity and high variation between samples over the sampling period. Nevertheless, the distribution of ASVs between seasons showed that the number of different ASVs was higher in spring and summer and lower in fall and winter (Fig. 4B). These observed trends are consistent with other studies showing seasonality in protist diversity (16, 61, 62). Phylogenetic analysis revealed that within each family, some clades of ASVs appeared to be either ubiquitous and found in all fractions, all sites, and all seasons, while other clades of ASVs appeared to be more restricted. The Tubulinea taxon is the best example to illustrate this, as it contains three completely different taxonomic groups of ASVs (Fig. S4). One of them tends to be ubiquitous, found in both fractions, at the three sites, and during the four seasons (clade 6). The other two clades (clades 7 and 8) appear to be more specialized, showing several differences that suggest two different lifestyles. A more resolutive sampling strategy may be required to fully capture the seasonal variation of the different taxonomic groups.

Biotic factors such as prey can also influence the composition of heterotrophic protist communities. Some prey have evolved the ability to resist predation using various extracellular and intracellular mechanisms, sometimes even killing them (13, 19, 29). F. Amaro et al. have shown that *Legionella pneumophila* can shape protist communities in microcosm experiments, with significant effects on the abundance of the phyla Cercozoa, Amoebozoa, and Heterolobosea (14). Here, we wondered whether three different strains of opportunistic vibrios with different anti-eukaryotic virulence mechanisms could influence the diversity of FLA communities. Surprisingly, the highest diversity was observed when *V. harveyi* Th15_O_A01 was used as a food source compared to the other strains, which showed three times more specific ASVs than *E. coli* SBS363 (Fig. 4B). On the contrary, *V. tasmaniensis* LGP32 and to a lesser extent *V. crassostreae* J2-9 were the most selective bacterial lawns as we found a lower amount of specific ASVs, suggesting that their anti-eukaryotic defense mechanisms could be efficient against a greater diversity of amoebozoa. We observed more generalist or more specialist clades of ASVs in most Amoebozoa families, with some ASVs that could grow with any of the prey, suggesting the existence of generalist species with a wide range of prey and habitats. However, their number was limited, and many ASVs had more specialized grazing capacity toward the different prey. Among them, the majority of Vannellidae and a large part of Tubulinea could not grow in the presence of LGP32. This is reminiscent of our previous study showing that *V. tasmaniensis* LGP32 is resistant to predation by *Vannella* sp. AP1411 (19). Paramoebidae was the family that contained the highest number of different ASVs, but their growth was particularly inhibited by J2-9. The contrasts in prey specificity of Vannellidae and Paramoebidae were functionally confirmed using *Vannella* sp. AP1411 and *Paramoeba atlantica* with both vibrios. These results emphasize that although intrageneric variations in predation capacity are observed, the resistance to grazing of some vibrios, such as *V. tasmaniensis* LGP32 and *V. crassostreae* J2-9, can affect larger taxonomic groups of Amoebozoa. Seasonal variations in the abundance of *V. tasmaniensis* and *V. crassostreae* in different habitats have been previously reported in the Thau lagoon (47). These *Vibrio* strains were found more abundant in the water column during the warmer seasons but restricted to the sediment during the colder seasons. Because Vannellidae are more abundant in the water column and Paramoebidae are more abundant in the sediment, the differences

in Amoebozoa diversity in the two different habitats could have opposite effects in shaping *Vibrio* communities and differentially affect the dynamics of different pathogens over the seasons. Similarly, *in vitro* functional grazing assays have been used by others to highlight differential predation capacity and prey specificity and to predict potential effects on bacterial communities in the plant rhizosphere (63). Interestingly, some interactions between vibrios and Paramoebidae have been reported previously; their role in *Vibrio* population dynamics and as a potential intracellular niche needs further investigation (34, 64).

In conclusion, our study provides a better understanding of Amoebozoa grazer communities in Mediterranean coastal environments and sets the stage for further in-depth functional studies between different Amoebozoa taxonomic groups and the bacterial communities present in these environments. By using different marine vibrios, we bring new evidence that Amoebozoa-*Vibrio* interactions are highly diverse and underline the need to further study Amoebozoa diversity and their role in *Vibrio* community dynamics and pathogen emergence.

## ACKNOWLEDGMENTS

We are grateful to Eve Toulza, Jérémie Vidal Dupiol, and Jean-Christophe Auguet for fruitful discussions and precious help in sequencing analysis. We thank Philippe Haffner and Marc Leroy for technical assistance. Sequencing was performed by the GENSEQ platform from UAR 2040 MEEB.

The present study was supported by the Ec2co-CNRS funded VibrAm project, by the UE funded project VIVALDI (H2020 program, no. 678589), by the EU funded EMBRC and by Ifremer, University of Montpellier and University of Perpignan via Domitia.

## AUTHOR AFFILIATIONS

[1]IHPE, University of Montpellier, CNRS, Ifremer, University of Perpignan Via Domitia, Montpellier, France
[2]SeBiMER Service de Bioinformatique de l'Ifremer, Ifremer, IRSI, Plouzané, France
[3]Sorbonne Universités, UPMC Univ Paris 06, CNRS, Observatoire Océanologique de Banyuls (OOB), Banyuls/Mer, France
[4]Laboratoire de Biodiversité et Biotechnologies Microbiennes (LBBM), Sorbonne Universités, UPMC Univ Paris 06, CNRS, Observatoire Océanologique, Banyuls/Mer, France
[5]MARBEC, University of Montpellier, CNRS, IRD, Ifremer, Montpellier, France

## AUTHOR ORCIDs

Angélique Perret http://orcid.org/0000-0001-8445-6424
Delphine Destoumieux-Garzón http://orcid.org/0000-0002-8793-9138
Guillaume M. Charrière http://orcid.org/0000-0002-4796-1488

## FUNDING

| Funder | Grant(s) | Author(s) |
| --- | --- | --- |
| Centre National de la Recherche Scientifique | INSU-Ec2co | Guillaume M. Charrière |
| Centre Méditerranéen de l'Environnement et de la Biodiversité | | Guillaume M. Charrière |
| Laboratoire d'Excellence TULIP | | Guillaume M. Charrière |
| Institut Français de Recherche pour l'Exploitation de la Mer | | Etienne Robino |

## AUTHOR CONTRIBUTIONS

Etienne Robino, Conceptualization, Data curation, Formal analysis, Investigation, Methodology, Visualization, Writing – original draft | Angélique Perret, Data curation,

Formal analysis, Investigation, Writing – review and editing | Cyril Noel, Formal analysis, Methodology, Software, Validation | Philippe Haffner, Investigation | Laurent Intertaglia, Investigation, Writing – review and editing | Marion Richard, Investigation, Writing – review and editing | Noémie Descamps, Investigation | Axelle Sellier, Investigation | Laura Onillon, Investigation | Philippe Lebaron, Resources | Delphine Destoumieux-Garzón, Supervision | Guillaume M. Charrière, Conceptualization, Data curation, Formal analysis, Funding acquisition, Investigation, Methodology, Supervision, Validation, Writing – original draft, Writing – review and editing

## DATA AVAILABILITY

Data are available under the BioProject PRJEB87851 on the ENA database https://www.ebi.ac.uk/.

## ADDITIONAL FILES

The following material is available online.

### Supplemental Material

**Supplemental material (Spectrum01138-25-s0001.pdf).** Fig. S1 to S5; Tables S2, S3, and S6.
**Table S1 (Spectrum01138-25-s0002.csv).** ASV *E. coli* list.
**Table S4 (Spectrum01138-25-s0003.xlsx).** ANCOM data.
**Table S5 (Spectrum01138-25-s0004.csv).** ASV *Vibrio* list.

### Open Peer Review

**PEER REVIEW HISTORY (review-history.pdf).** An accounting of the reviewer comments and feedback.

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
