## [Reviewer comments · Microbiology Spectrum]

Microbiology Spectrum

Diversity and varying predation capacities of culturable Amoebozoa against opportunistic vibrios in contrasting environments.

Etienne Robino, Angelique Perret, cyril noel, Philippe Haffner, Laurent Intertaglia, Marion Richard, Noemie Descamps, Axelle Sellier, Laura Onillon, Philippe Lebaron, Delphine Destoumieux-Garzon, and Guillaume Charriere

Corresponding Author(s): Guillaume Charriere, IHPE, University of Montpellier

Review Timeline:

Submission Date:	April 15, 2025
Editorial Decision:	July 22, 2025
Revision Received:	October 31, 2025
Accepted:	November 14, 2025

Editor: Erik Hom

Reviewer(s): Disclosure of reviewer identity is with reference to reviewer comments included in decision letter(s). The following individuals involved in review of your submission have agreed to reveal their identity: Christopher Aaron Rice (Reviewer #1); Marco Gabrielli (Reviewer #2)

Transaction Report:

DOI: <https://doi.org/10.1128/spectrum.01138-25>

Re: Spectrum01138-25 (Diversity and varying predation capacities of Amoebozoa against opportunistic vibrios in contrasting Mediterranean coastal environments.)

Dear Dr. Guillaume M Charriere:

Thank you for the privilege of reviewing your work, and again for your patience as we had to wait on qualified reviewer candidates to agree to review your work.

Below you will find my comments, instructions from the Spectrum editorial office, and the reviewer comments.

Please make the adjustments requested by Reviewer #2 to control for confounding factors in your analyses. Also, please make sure that the BioProject ID for your data is functional.

Revision Guidelines

Sincerely,
Erik Hom
Editor
Microbiology Spectrum

Reviewer #1 (Public repository details (Required)):

The authors state that they have deposited their data in the

"Data Availability

Data is available under the BioProject PRJEB87851 on ENA database <https://www.ebi.ac.uk/>."

Reviewer #1 (Comments for the Author):

Please see major and minor comments attached.

Reviewer #2 (Public repository details (Required)):

The manuscript includes the analysis of sequencing data. A BioProject ID is mentioned but it was not possible to find.

Reviewer #2 (Comments for the Author):

General comments:

The manuscripts describes the diversity of culturable Amoebozoa in the water phase and sediments in three marine sampling locations. The Authors describe the richness and beta diversity among those communities and then assess the effect of different preys on eukaryotic survival highlighting how different clades respond differently to the presence of distinct Vibrio.

It would be nice to see more statistical validation of some of the qualitative claims made throughout the manuscript, as well as removing the potential effect of variable sequencing depth on alpha and beta diversity.

Also the data does not seem yet to have been uploaded to the mentioned BioProject.

I would ask the Authors to add line numbers to ease reviews in making specific comments.

Specific comments:

- I would ask the Authors to add the word "culturable" in the title as all the work is based on 18S sequences of amoebae which could be grow on the bacteria lawns

Introduction:

- "eukaryotic" is spelled as "eucaryotic" in a couple of instances throughout the text

- The acronym "POMS" has not been previously defined

- I would ask the Authors to clarify why the three sampling sites are "contrasting".

Materials and Methods:

- I would encourage the Authors to describe how they collected the amoebae from the bacterial lawn in the sampling section

- A repeated sentence about the SAMBA pipeline is present in the "18S barcoding data processing" section. Furthermore, I suggest the Authors to merge this section with the one titled "Amoebozoa diversity analysis on different nutritive sources" as this provides detailed information about the data processing.

- It seems like the Authors analyzed separately the data from the Ecoli lawn and the other bacteria. I think this is not the best strategy as it would make harder comparing the effect at the ASV level. I would encourage the Authors to analyze all the samples together or better explain the rational behind this choice in the manuscript.

- Given the influence of phylogeny on UniFrac distances, I would ask the Authors whether the estimated phylogenetic tree has been visually assessed the plausibility of the tree. For exxample, in Figure 5 several very long branches exist and I wonder what are the ASVs producing them as well as how much they influence the analysis

- In general I would ask the Authors to make clearer the distinction between the when they refer to the whole time series or the samples related to the oyster mortalit events to ease readers' understanding.

Results:

- The first section of the results summarises the content of the Meterial and Methods section. I would ask the Authors to remove or drastically shorten this.

- The Authors mention that water samples seem more grouped than water ones. I would suggest them to use an appropriate statistical test (e.g., permdisp) to assess that.

- The Authors either do not rarefy their sequences or use DESeq2 normalization for their beta diversity. However, these are approaches that are known to potentially influence the results (10.1128/msphere.00354-23). I would strongly encourage the Authors to rarefy their samples to an equal sequencing depth to compare alpha and beta diversity.

- ANCOM analysis refers to differential abundance. Given the fact, that culturing was performed for two weeks before analysis, it is not necessarily true that the relative abundances measured are representatives of the ones in-situ due to, for example, different growth rates of different clades in the experimental conditions used. Given the experimental procedure used, I would strongly advise the Authors to rely on relative abundance metrics during the analysys of this dataset.

- Figures 1D-E seem to show differential presence of given families across different sites. I would encourage the Authors to test this.

- I find a bit odd to refer to Figure 3, before having discussed Figure 2.

- Te Authors claim that the taxonomic affiliations of specific ASVs in summer and spring are similar. I would strongly encourage to check this statistically.

- I would strongly encourage the Authors to provide some statistical support on their interpretation of the trees results.

- The Authors state that there is no difference in beta diversity among the plates with different bacteria, but that 58.7% of ASVs are present in a single lawn type. I would strongly ask them to explain this apparent contradiction.
- When comparing the relative abundance of Vannellidae and Paramoebidae it is not clear to me what the Authors mean when they say "total abundance". Also, while in the text the analysis seems to be focused on comparing the two clades, the image is more focused on comparing the effect of the lawn on the single clade. I would ask the Authors to clarify.

This study looks at the time and species distribution of Amoebozoa from the Thau lagoon and Mediterranean Sea to help understand the diversity between three sampling sites. This study assessed the Amplicon Sequence Variants (ASVs) of amoeba from the water column and sediment over several months and validated presence of particular amoeba from bacterial lawn plates of *E. coli* and *Vibrio* strains over these time periods.

Major Comments

Reference to Figures are out of order. Please refer to figure 2 before figure 3 in the results section of the text.

Minor Comments

Introduction

You define “free-living amoeba (FLA)” then in the very next sentence you do not abbreviate this.

Sentence needs updated “amoebzoa is the only group that contains only FLAs” Naegleria is a FLA and is in the Excavata phylum. This is not true sentence, remove it.

Suggest changing these sentences “Free-living amoeba feed on various microorganisms and digest them by phagocytosis. They can feed on bacteria, yeasts, fungi, algae or other protists”. It could read.

“FLA feed on various microorganisms, including bacteria, yeasts, fungi, algae or other protists, by phagocytosis and digest them in digestive vacuoles (Refs).

Define OsHV-1.

Sentence with reference (lee et al., 2013) has a blue period at the end of the sentence.

Define POMS, you refer to Pacific oyster syndrome mortality above. Change this to POMS and abbreviate it.

Materials and methods

Please use “ $\mu\text{g}/\text{mL}$ ” also in the grazing assay method for concentrations of cell/mL.

You use “-80°C” then use “minus 80°C”, use one for consistency.

Data project could not be found using the link and code. This is usually typically until it is published and then released.

Define ASVs.

Repeated text “which consists of only FLA” before (Gillou et al., 2012 reference)

Results

Please include a space between “these conditions.After 15 days of”

Please include a reference to the 18S universal primers or include these in a supplemental table.

In this section “Seasonal variations in Amoebozoa diversity occur at the subgenus level.” Please check there isn’t a double space before “Paramoebidae”.

Figures

Fig 3. – you do not reference what the numbers 1-14 mean. In another figure it relates to clades. Please specify in the figure legend and or main text.

Fig 7. - Please show which days actually show statistical significance. As not all days (D0-D1, would be significant).

Supp. Fig. 1 – Please colour code and add appropriate annotations for each of the sites to align with the nomenclature used throughout all main figures and text.

Reviewer comments :

Reviewer #1 (Public repository details (Required)):

The authors state that they have deposited their data in the "Data Availability Data is available under the BioProject PRJEB87851 on ENA database <https://www.ebi.ac.uk/>."

Reviewer #1 (Comments for the Author):

Please see major and minor comments attached.

This study looks at the time and species distribution of Amoebozoa from the Thau lagoon and Mediterranean Sea to help understand the diversity between three sampling sites. This study assessed the Amplicon Sequence Variants (ASVs) of amoeba from the water column and sediment over several months and validated presence of particular amoeba from bacterial lawn plates of *E.coli* and *Vibrio* strains over these time periods.

Major Comments

Reference to Figures are out of order. Please refer to figure 2 before figure 3 in the results section of the text.

Thank you for helping us to improve our manuscript. This has been corrected.

Minor Comments

Introduction

You define "free-living amoeba (FLA)" then in the very next sentence you do not abbreviate this.

Done.

Sentence needs updated "amoebzoa is the only group that contains only FLAs" Naegleria is a FLA and is in the Excavata phylum. This is not true sentence, remove it.

Done.

Suggest changing these sentences "Free-living amoeba feed on various microorganisms and digest them by phagocytosis. They can feed on bacteria, yeasts, fungi, algae or other protists". It could read.

"FLA feed on various microorganisms, including bacteria, yeasts, fungi, algae or other protists, by phagocytosis and digest them in digestive vacuoles (Refs).

Done.

Define OsHV-1

Replaced by "viral" as specification of this particular virus is not relevant for this work.

Sentence with reference (lee et al., 2013) has a blue period at the end of the sentence.

Done.

Define POMS, you refer to Pacific oyster syndrome mortality above. Change this to POMS and abbreviate it.

Done.

Materials and methods

Please use “ $\mu\text{g/mL}$ ” also in the grazing assay method for concentrations of cell/mL.

Done.

You use “ -80°C ” then use “minus 80°C ”, use one for consistency.

Done.

Data project could not be found using the link and code. This is usually typically until it is published and then released.

Done.

Define ASVs.

Done.

Repeated text “which consists of only FLA” before (Gillou et al., 2012 reference)

Removed.

Results

Please include a space between “these conditions.After 15 days of”

Done.

Please include a reference to the 18S universal primers or include these in a supplemental table.

Done, (Stoeck et al., 2010).

In this section “Seasonal variations in Amoebozoa diversity occur at the subgenus level.”

Please

check there isn’t a double space before “Paramoebidae”.

Done

Figures

Fig 3. – you do not reference what the numbers 1-14 mean. In another figure it relates to clades.

Please specify in the figure legend and or main text.

Done

Fig 7. - Please show which days actually show statistical significance. As not all days (D0-D1, would be significant).

Done

Supp. Fig. 1 – Please color code and add appropriate annotations for each of the sites to align with the nomenclature used throughout all main figures and text.

Done.

Reviewer #2 (Public repository details (Required)):

The manuscript includes the analysis of sequencing data. A BioProject ID is mentioned but it was not possible to find.

[The BioProject is now publicly available.](https://www.ebi.ac.uk/ena/browser/view/PRJEB87851)
<https://www.ebi.ac.uk/ena/browser/view/PRJEB87851>

Reviewer #2 (Comments for the Author):

General comments:

The manuscript describes the diversity of culturable Amoebozoa in the water phase and sediments in three marine sampling locations. The Authors describe the richness and beta diversity among those communities and then assess the effect of different preys on eukaryotic survival highlighting how different clades respond differently to the presence of distinct *Vibrio*.

It would be nice to see more statistical validation of some of the qualitative claims made throughout the manuscript, as well as removing the potential effect of variable sequencing depth on alpha and beta diversity.

Thank you for your recommendations, we ran additional statistical analyses (see below) and we were more cautious and precise in the description of the data and the interpretations and conclusions in the revised version of the text.

Also, the data does not seem yet to have been uploaded to the mentioned BioProject.

[The BioProject is now publicly available.](https://www.ebi.ac.uk/ena/browser/view/PRJEB87851)
<https://www.ebi.ac.uk/ena/browser/view/PRJEB87851>

I would ask the Authors to add line numbers to ease reviews in making specific comments.

We apologize for this mistake, line numbers have been included in the revised manuscript.

Specific comments:

- I would ask the Authors to add the word "culturable" in the title as all the work is based on 18S sequences of amoebae which could be grow on the bacteria lawns

We agree, this was done.

Introduction:

- "eukaryotic" is spelled as "eucaryotic" in a couple of instances throughout the text

Done.

- The acronym "POMS" has not been previously defined

Done.

- I would ask the Authors to clarify why the three sampling sites are "contrasting".

A more detailed description of sampling sites differences was added at the beginning of the results section: "The Thau lagoon, which is used for oyster farming and where oyster mortality due to POMS occurs annually, it represents a semi-enclosed coastal system with restricted seawater renewal and substantial freshwater inputs from watersheds, resulting in salinity variations (Richard et al., 2021), but also two other sites, in the Mediterranean Sea outside the port of Sète, an open sea area where there is no aquaculture activities, and near the marine protected area of Banyuls-sur-Mer which is much farther from the two other sites, with restricted human activities to protect the marine biodiversity. »

Materials and Methods:

- I would encourage the Authors to describe how they collected the amoebae from the bacterial lawn in the sampling section.

This was added in the methods section, see line 175 to 179.

- A repeated sentence about the SAMBA pipeline is present in the "18S barcoding data processing" section. Furthermore, I suggest the Authors to merge this section with the one titled "Amoebozoa diversity analysis on different nutritive sources" as this provides detailed information about the data processing.

The modifications have been made accordingly.

- It seems like the Authors analyzed separately the data from the E. coli lawn and the other bacteria. I think this is not the best strategy as it would make harder comparing the effect at the ASV level. I would encourage the Authors to analyze all the samples together or better explain the rationale behind this choice in the manuscript.

We did not perform FLA enrichment on all four types of bacterial lawns every month of the sampling campaign; this was only performed on four different dates. Therefore, to analyze environmental variations and seasonal effects on FLA communities, we chose to analyze all the monthly samples from E. coli lawns only.

Then, to investigate the specific effects of using different bacterial lawns, all samples acquired during the four specific months were compared, including the E. coli samples from those months. As all the data analysis was based on ASVs, ASVs present in the different samples remain the same, and can be compared and are not relative to the diversity of reads sequences in the set of data, as opposite to OTUs.

- Given the influence of phylogeny on UniFrac distances, I would ask the Authors whether the estimated phylogenetic tree has been visually assessed the plausibility of the tree. For example, in Figure 5 several very long branches exist and I wonder what are the ASVs producing them as well as how much they influence the analysis

We thank the reviewer for this insightful comment. We acknowledge that the accuracy of the inferred phylogenetic tree can influence UniFrac-based analyses and that the presence of long branches may raise concerns regarding alignment or taxonomic placement. The phylogenetic tree generated here are based on relative distances and should therefore be interpreted with caution. Nevertheless, we have visually inspected the tree and compared the positions of representative ASVs with reference sequences from known Amoebozoa families and genera. Overall, the topology appears plausible and biologically consistent with established classifications.

- In general, I would ask the Authors to make clearer the distinction between the when they refer to the whole time series or the samples related to the oyster mortality events to ease readers understanding.

Done

Results:

- The first section of the results summarizes the content of the Material and Methods section. I would ask the Authors to remove or drastically shorten this.

Done

- The Authors mention that water samples seem more grouped than sediment ones. I would suggest them to use an appropriate statistical test (e.g., permdisp) to assess that.

We thank the reviewer for this valuable suggestion. We agree that our initial statement regarding the apparent grouping of water versus sediment samples required statistical support. Following the reviewer's recommendation, we conducted a permutation test for multivariate dispersion (PERMDISP), which confirmed that water samples exhibited significantly lower dispersion compared to sediment samples ($p = 0.037$). This analysis has been added to the revised manuscript (Methods section, lines 263-264; Results section, lines 326-327).

- The Authors either do not rarefy their sequences or use DESeq2 normalization for their beta diversity. However, these are approaches that are known to potentially influence the results (10.1128/msphere.00354-23). I would strongly encourage the Authors to rarefy their samples to an equal sequencing depth to compare alpha and beta diversity.

We appreciate the reviewer's comment regarding sequence normalization. Normalization is indeed an important step in metabarcoding data analysis. This is why, in this study and particularly using the SAMBA workflow, three different normalization methods were tested: rarefaction (1000 iterations), DESeq2, and CSS, as suggested by McMurdie & Holmes (2014). After a thorough comparison of these approaches, we selected DESeq2 normalization, as it provided the best fit for our data. Rarefaction curves revealed a high variability in sequencing depth among samples, leading to a substantial loss of information when

subsampling to a uniform depth. In contrast, DESeq2 normalization preserved more of the underlying community structure and provided more stable diversity estimates across replicates. To address the reviewer's concern, we have now included the rarefaction curves (Figure S5) to illustrate the variability in sequencing depth and to justify our normalization choice.

- ANCOM analysis refers to differential abundance. Given the fact, that culturing was performed for two weeks before analysis, it is not necessarily true that the relative abundances measured are representatives of the ones in-situ due to, for example, different growth rates of different clades in the experimental conditions used. Given the experimental procedure used, I would strongly advise the Authors to rely on relative abundance metrics during the analyses of this dataset.

We agree that the use of differential or absolute abundance analyses is particularly challenging in our experimental context, since the two-week cultivation step may have altered the relative abundances of taxa due to differential growth rates under the culture conditions. For this reason, our initial approach focused primarily on presence/absence data and qualitative comparisons between samples, which we considered more reliable indicators of community composition given the potential biases introduced by culturing. Nevertheless, in line with the reviewer's suggestion, we have now included additional analyses based on relative abundance metrics (using the *indicspecies* function from the *vegan* package). These complementary results highlight a high heterogeneity among samples from the same site, but overall show trends consistent with the ANCOM-based findings, supporting the robustness of our main conclusions. This addition and its interpretation are now described in the revised manuscript (Methods section, lines 279–282; Results section, lines 413–442).

- Figures 1D-E seem to show differential presence of given families across different sites. I would encourage the Authors to test this.

Indeed, we complemented our analyses by adding Ancom and *indicspecies* (*Vegan* package) as discuss above. See Results section and supplemental figures and tables (as above).

- I find a bit odd to refer to Figure 3, before having discussed Figure 2.

This was corrected.

- The Authors claim that the taxonomic affiliations of specific ASVs in summer and spring are similar. I would strongly encourage to check this statistically.

We apologize for the poor phrasing of this sentence. The text has been revised for greater clarity and accuracy.

- I would strongly encourage the Authors to provide some statistical support on their interpretation of the trees results.

The phylogenetic trees are relative trees, in order to better analyze ASVs diversity in the datasets. Still, we verified representative ASVs of the different clades with reference sequences from public databases (SILVA or PR2), and the affiliations appeared consistent. Clades were identified both based on phylogeny relationships, branch length, as well as

environmental data. This was done for ease of result description and discussion. Molecular phylogeny and taxonomic assignment of Amoebozoan is still not optimal due to the lack of well curated reference databases, like for most unicellular eukaryotes.

- The Authors state that there is no difference in beta diversity among the plates with different bacteria, but that 58.7% of ASVs are present in a single lawn type. I would strongly ask them to explain this apparent contradiction.

In order to investigate further this apparent discrepancy, we performed PERMANOVA analysis which showed no significant difference in beta diversity among the plates with different bacterial lawns, indicating that overall community structures are not statistically different. However, it is true that 58.7% of ASVs were detected in only one lawn type. This pattern primarily reflects the presence of many rare ASVs occurring at very low abundances or as singletons, which do not significantly influence community-level dissimilarity metrics used in the beta diversity analysis. We have clarified this point in the revised manuscript (Results section, lines 576–578).

- When comparing the relative abundance of Vannellidae and Paramoebidae it is not clear to me what the Authors mean when they say "total abundance". Also, while in the text the analysis seems to be focused on comparing the two clades, the image is more focused on comparing the effect of the lawn on the single clade. I would ask the Authors to clarify.

We apologize for this confusion. The text and phrasing has been revised for greater clarity and accuracy.

Re: Spectrum01138-25R1 (Diversity and varying predation capacities of culturable Amoebozoa against opportunistic vibrios in contrasting environments.)

Dear Dr. Guillaume M Charriere:

Your manuscript has been accepted, and I am forwarding it to the ASM production staff for publication. Your paper will first be checked to make sure all elements meet the technical requirements. ASM staff will contact you if anything needs to be revised before copyediting and production can begin. Otherwise, you will be notified when your proofs are ready to be viewed.

Sincerely,
Erik Hom
Editor
Microbiology Spectrum

Reviewer #2 (Comments for the Author):

I thanks the Authors for the additional work spend on improving the manuscript. I have no further comments.